# Damage Localization of Composites Based on Difference Signal and Lamb Wave Tomography

**DOI:** 10.3390/ma13010218

**Published:** 2020-01-04

**Authors:** Chenhui Su, Mingshun Jiang, Jianying Liang, Aiqin Tian, Lin Sun, Lei Zhang, Faye Zhang, Qingmei Sui

**Affiliations:** 1School of Control Science and Engineering, Shandong University, Jinan 250061, China; suchenhui2010@163.com (C.S.); drleizhang@sdu.edu.cn (L.Z.); zhangfaye@sdu.edu.cn (F.Z.); qmsui@sdu.edu.cn (Q.S.); 2CRRC Qingdao Sifang Co. Ltd., Qingdao 266111, China; sfliangjianying@sina.com (J.L.); sftianaiqin@sina.com (A.T.); sfsunlin@aliyun.com (L.S.)

**Keywords:** composite materials, damage, identification, lamb wave tomography

## Abstract

In order to deal with the problem of composite damage location, an imaging technique based on differential signal and Lamb wave tomography was proposed. Firstly, the feasibility of the technique put forward was verified by simulation. In this process, the composite model was regularly set down by the circular sensor array, with each sensor acting as an actuator in sequence to generate Lamb waves. Apart from that, other sensors were used to collect response signals. With regard to the damage factor, it was mainly determined by the difference between the damage signal and the non-damage signal. The probabilistic imaging algorithm was employed to carry out damage location imaging. Then, experiments were carried out so as to study the selected composite plate. Finally, the tentative outcomes have illustrated that the maximum error of damage imaging position was 7.07 mm. The relative error was 1.6%. In addition, the method has the characteristics of simple calculation and high imaging efficiency. Therefore, it has large technical potential and wide applications in the damage location and damage recognition for composite material.

## 1. Introduction

Carbon fiber reinforced polymer (CFRP) is well-known for its advantages of light weight, high strength and strong design ability and therefore it is widely applied in the fields of aviation, aerospace, and high-speed railway etc. [1,2,3]. While, composite structures of carbon fiber are susceptible to be damaged by exterior shocks and concentrated loads during manufacture or service, and serious accidents may occur consequently if they fail to be inspected and maintained in time [4]. Therefore, developing an accurate damage location method is quite necessary in guaranteeing the safety of composite structures.

Lamb wave has aroused wide concern for their three characteristics as long distance of propagation, low cost and good sensitivity to various defects, which enables them to be the focus of attention in the field of non-destructive testing of composite [5,6]. An increasing number of methods for time-of-flight (TOF) calculation have been well developed as a variety of researchers have adopted TOF techniques to determine the exact location of damage in composite plates. For instance, Xu [7] proposed a method which integrated sparse reconstruction with delay-and-sum (DAS) for the inspection of high-resolution Lamb wave, and the method of DAS imaging and characteristic signal were also combined with each other so that it can be applied to the damage imaging of CFRP. Zhang [8] obtained damage imaging of composite structures by applying the algorithm of Lamb wave probabilistic imaging. In addition, Huang [9] proposed an improved method of time reversal according to the time invertibility of Lamb wave. In accordance with the time difference between wave velocity and scattering signal, the damage location of composites can be achieved. However, since it is not easy to clearly understand the allotropy of the composite structure and the propagation of the Lamb wave, it is quite difficult to identify the specific law only using the index of the signal [10,11,12]. Thereby, many scholars carried out relevant research to identify the damage position in composite material based on pattern recognition. For instance, Su [13] detected the actual delamination of composite laminates by using an artificial neural network, which proved the feasibility of the method used for pattern recognition in damage detection of composite materials. De [14] detected the damage location and degree of composite plates by using the combination of an artificial neural network and the method of probabilistic ellipse. Sun [15] proposed a method of damage quantification using Lamb wave based on least squares support vector machine and a genetic algorithm. Yet, this technique with pattern recognition as the basis is in the need of a large amount of sample data, which is regarded as a significant factor that restricts its rapid development.

Therefore, scholars have studied Lamb wave tomography technology. It only considers the relationship between the measured signal and the baseline signal to define the damage index. It does not need parameters such as wave velocity to realize damage imaging. Damage indication based on a single damage index value demonstrated its advantages in other methods such as vibro-acoustic modulation technique which facilitates interpretation of damage in the structures as well as monitoring of the damage evolution [16,17,18]. Liu [4] uses Lyapunov to characterize the relationship between the measured signal and the baseline signal to define the damage index, and realizes the imaging of delamination damage of composite plates. Cai [19] used a correlation dimension to define damage index to detect composite material damage. In the process of correlation dimension calculation, it involves time delay, embedding dimension, the solution of curve from LnC(r)~Lnr, and the slope obtained by linear fitting. Zhou [20] used fractal dimension to define the damage index to realize damage imaging of composite materials. In practical calculation, a series of square grids are used to cover the signal in the scale-free area of the calculation sequence, and the effective number of covering grids is obtained. Finally, the slope of the fitting line is obtained by the least square method, which is the fractal box dimension of the signal. Xu [21] and Sheen B [22] use the statistical characteristics of signals to define the damage index to achieve structural damage imaging. These methods can identify the damage location of composites. This statistical characteristic involves the direct time of the signal and the time window of the signal. However, in these methods, the calculation of damage index involves complex mathematical problems and is inefficient, which is not conducive to rapid imaging of structural damage.

In order to deal with the above problems, a new technique of damage location for composite materials was put forward according to the differential signal and Lamb wave tomography. More than that, the propagation characteristics and damage mechanism of Lamb wave in composite materials were simulated and analyzed. Apart from that, the feasibility of this method was also verified by simulation and experiment. During the process, the difference between the signals before and after the damage would be identified by the method of differential signal, followed by the calculation of the damage index. Finally, damage location imaging was achieved.

## 2. The Mechanism of Lamb Wave Damage Detection

It is known that the comprehension over the propagation process of Lamb wave in composite materials is a prerequisite for experimental study. Therefore, a composite material model with a dimension of 600 mm × 600 mm × 2 mm and a laying mode of (0°/90°)_8_ was established by using ABAQUS software (6.13, DA SIMULIA, France). Table 1 shows the parameters of material.

In terms of the position of actuators and sensors on the CFRP, they are shown in Figure 1a. The actuator was responsible for sending out Lamb waves so as to propagate in the board, and the sensor could be employed to receive the response signal. In order to ensure the accuracy and efficiency of the measurement, the grid unit size is 1 mm × 1 mm; the element type is SC8R; the finite element model consists of 720,000 continuous shell elements. On the basis of Formula (1) [23], the Lamb wave signal can be generated with a central frequency of 50 kHz and it was loaded on the actuator. The domain waveforms of time and frequency are illustrated in Figure 1b, and the standardized amplitude is shown in the ordinate.
(1)$$A=\frac{1}{2}\left[1-\cos\left(\frac{2\pi f_{c}t}{n}\right)\sin\left(2\pi f_{c}t\right)\right]$$
*f_c_* refers to the frequency; *n* denotes the number of cycles in the signal window (*n* = 5); *t* represents the duration of wave propagation.

Figure 2a,b was respectively demonstrated the process of propagation of Lamb wave in undamaged and damaged CFRP. The results showed that the damage was caused by a hole with 20 mm diameter. In the simulation analysis, the A0 wave generated by the out-of-plane force is extracted.

Figure 2 shows that the wave front is not a standard circular wave front but a wave front that is approximate to rhombus, which indicates each direction of composite plate is different. Moreover, the wave propagation velocity in the composite material was subject to the impact of the laying mode, and the wave propagation velocities in different directions were also different from each other. From Figure 2b, it can also be observed that Lamb waves would generate scattering waves when they suffer from damage, which would consequently enable the amplitude of the direct wave response signal obtained by the sensor to change compared with that of the undamaged signal. As a matter of fact, change of this type could be represented by the signal difference existing before and after the damage, and the location of the damage could be identified through the employment of Lamb wave tomography.

## 3. Principle of Lamb Wave Tomography

Lamb wave tomography is actually a method obtained on the basis of correlation analysis. It is able to recognize the damage position through the calculation conducted for the signal change before and after damage, in which parameters such as wave velocity are usually not required. In the meanwhile, neither the analysis of the complex multimode propagation characteristics of Lamb waves nor understanding and modelling the characteristics of materials or structures is required by the algorithm. The technology includes two parts: signal comparison and image reconstruction. As for the signal comparison, the difference signal was obtained by the subtraction carried for the signals obtained before and after damage, as shown in Formula (2). For the reason that the signal is discrete, it could be expressed in Formula (3) as follows. According to Formula (4), the sum of squares of the difference signal was obtained, which is expressed as damage factor *DF*_1_.
(2)Signal(t)=y_und(t)−y_d(t)
(3)$$Signal\left(t\right)=[x_{1},x_{2},x_{3}\ldots x_{n}]$$
(4)$$DF_{1}=\sum_{i=1}^{n}x_{i}^{2}$$
*Y_und*(*t*) and *y_d*(*t*) refer to the Lamb wave response signals collected by the sensor in the condition of no damage or damage was available respectively. Based on the *DF1* value corresponding to each sensor path, the probability distribution of damage obtained from adjacent regions was then reconstructed. It could be observed from the reconstructed image that each *DF*_1_ value was arranged on an elliptic surface, and the exciting end I and the receiving end J in corresponding sensing path were the two focal points of the ellipse. The definition for the spatial distribution function of *DF*_1_ value was shown as follows [24]:(5)$$\left\{ \begin{array}{ll} S_{ij}(x,y)=\frac{\beta -R_{i}{}_{j}(x,y)}{1-\beta } & \beta >R_{ij}(x,y)\\ S_{ij}(x,y)=0 & \beta \leq R_{ij}(x,y) \end{array}\right.$$
where, in the formula: $R_{ij}\left(x,y\right)$ refers to the ratio of the sum of the distance from the point (x,y) to the actuator $\left(x_{ik},y_{ik}\right)$ and the sensor $\left(x_{jk},y_{jk}\right)$ to the length of the sensing path; β denotes the shape factor, which controls the size of the ellipse, and β is greater than 1. $R_{ij}\left(x,y\right)$ can be measured by the followed equation:(6)$$\begin{array}{ll} R_{ij}\left(x,y\right) & =\frac{\sqrt{{\left(x-x_{ik}\right)}^{2}+{\left(y-y_{ik}\right)}^{2}}}{\sqrt{{\left(x_{ik}-x_{jk}\right)}^{2}}+\sqrt{{\left(y_{ik}-y_{jk}\right)}^{2}}}\\ & +\frac{\sqrt{{\left(x-x_{jk}\right)}^{2}+{\left(y-y_{jk}\right)}^{2}}}{\sqrt{{\left(x_{ik}-x_{jk}\right)}^{2}}+\sqrt{{\left(y_{ik}-y_{jk}\right)}^{2}}} \end{array}$$

In order to locate the damage with accuracy, the superimposition of all the perceptual paths corresponding by the maps of probability distribution is served to obtain the damage probability distribution of any point (*x*, *y*) in the regions of detection with N perceptual path [25]:(7)$$P(x,y)=\sum_{i=1}^{N-1}\sum_{j=i+1}^{N}DF_{1}*S_{ij}(x,y)$$

Finally, the signals obtained before and after the damage were employed so as to determine the damage factor, and the algorithm of probability imaging was used to realize the imaging of damage location. Figure 3 is a flow chart for damage location and imaging identification in composite materials.

The method can be specifically divided into following six steps:

The first step refers to the experiments for damage detection of Lamb wave were conducted by the means of simulation and experiment;

The second step is about extracting data from simulation through the employment of the simulation software and denoising the data from the experiment;

The third step is to calculate the damage factor based on the signals obtained from before and after damage according to Formula (4).

The fourth step refers to calculating the probability of damage according to Formula (7).

The fifth step is to scan the next node and Step 3–Step 4.

The final step is to explore the point with the maximum probability of damage on the basis of the realized damage imaging.

## 4. Numerical Simulation

For the purpose of verifying the feasibility of the method proposed, it was necessary to conduct the numerical simulation experiments.

The simulation was conducted with the second part of the paper as the basis, with a circular array of 12 sensors made of composite material was regularly set down in the center of the model. The array with 30 cm diameter and the damage was achieved by a hole with 20 mm diameter, with the location of coordinates (300, 375). With regard to the sensor position and damage coordinates of schematic diagram are shown in Figure 4. The parameters and excitation signals of the composites described in this part were the same as those in the second part. First, the sensor S1 and the remaining sensors were used to excite Lamb wave signals and receive lamb response signals of the structure, respectively. Then, the clockwise rotation sensor and other sensors were used for stimulating the Lamb wave signal and then collecting the response signal of Lamb wave, respectively, until all the sensors excitatory Lamb wave signal for one time. It should be noted that the first step of simulation was collecting the lamb signal in the non-destructive state, and then collecting the signal in the damaged state.

Figure 5a,b illustrate that sensor S7 and sensor S9 are responsible for receiving the Lamb wave signals sent by sensor S1 and sensor S5 respectively. The blue line is about the signal collected in the undamaged state, and the red line regards as the signal collected in the damaged state. When it was identified that the damage was at the point (300, 375), the direct wave signal received by the sensor became weakened due to the scattering of the direct wave signal as the damage located in the S1–S7 sensor channel. Since the S5–S9 sensor channel was free from being damaged, the direct wave signals were basically the same with each other. Additionally, as the scattering of signals would be given rise to the damage of composite materials, thus the amplitude of signals would be reduced accordingly. Therefore, the calculation of the damage factor could be carried out according to the signal difference before and after damage. The pink signal in Figure 5 denotes the difference signal and it could be seen that the damage would cause a larger difference signal.

On the basis of Formula (4), the measurement for the damage factors of each channel signal was carried out and then the factors were put into Formula (7) for image processing of the damage. The outcomes obtained from the composite damage imaging are illustrated in Figure 6. The coordinates in the picture refers to pixels, with each pixel value of 0.1mm. The result of threshold damage was located in the upper right corner.

The coordinates of the damage location is (300.1, 375.1). In order to assess the impact of positioning, it is advised to add the parameter of radial error into the calculation [26,27]:(8)$$e=\sqrt{{\left(x_{r}-x_{p}\right)}^{2}+{\left(y_{r}-y_{p}\right)}^{2}}$$
where, in the equation, e states the radial error; ($x_{r}$,$y_{r}$) represents the coordinate position of imaging damage; ($x_{p}$,$y_{p}$) signifies the coordinate of actual position. The result of the error of damage of imaging location could be calculated as 0.14 mm.

At the same time, the relative error is used to evaluate the damage location effect.
(9)$$\delta =\frac{\max(\Delta l_{H},\Delta l_{v})}{L}\times 100\%$$
where $\Delta l_{H}$ is the distance from experimental result to actual damage location in horizontal direction, and $\Delta l_{V}$ is the distance in vertical direction, $\max(\Delta l_{H},\Delta l_{v})$ is the maximum value of $\Delta l_{H}$ and $\Delta l_{V}$. *l* is length of the total sensors array in horizontal or vertical direction. The calculated relative error is 0.03%.

## 5. System Construction and Experiments

The geometric parameters of the composite material and the position of the sensor employed in the part of experiment were consistent with those in the simulation part. A total of 12 piezoelectric sensors were used in the process. The experimental system consists of an arbitrary function generator (Rigol DG5252), a linear wideband power amplifier (Krohn-Hite 7602M), a multi-channel oscilloscope (Tektronix MDO 4034B-3), and a computer, as identified in Figure 7.

Firstly, the function generator modulates the excitatory signal with a central frequency of 50 KHz through the Hanning window, and then loaded the Lamb wave signal onto the S1 sensor with the assistance of the amplifier. In the part of response signal, it was collected by other sensors using oscilloscope, and the sampling frequency was 10 MHz. The sensor was excited clockwise, and it was received by other sensors. Firstly, the response signal without damage was collected, which was followed by the collection of response signal with damage. The damage could be achieved under the change of local strain of the structure with mass blocks. Then, damage was realized at the coordinate position (315, 375) by pasting a mass block with a dimension of 30 mm × 10 mm × 40 mm.

It was inevitable that Lamb wave would be affected by noise in collecting signals. In order to remove the noise in the collection of the response signal, the wavelet transform technique which is known for excellent denoising performance was adopted to filtrate the noise of signals [28,29]. Figure 8 shows the original signal and the denoised signal.

Figure 9a,b show that S7 and sensor S9 were used to receive the excitatory signals sent by sensor S1 and sensor S5, respectively. It was consistent with the signal diagram in the part of simulation. The signal received by the sensor would be changed due to the existence of damage. The change could be shown by the signal difference before and after damage.

According to Formula (4), the damage factors of each channel signal was measured first and then the results were put into Formula (7) to carry out image processing for the damage. The results of composite damage imaging are shown in Figure 10, with the damage coordinate of (310, 380). According to Formula (8), the radial error obtained was 7.07 mm. According to Formula (9), the relative error is 1.6%.

For the aim of illustrating the advantages of the proposed method, it was compared with the existing methods of Lamb wave tomography. The calculation conducted for the damage factors of these methods was carried out based on energy and fractal theory, respectively.

The imaging and positioning structures of these three algorithms are shown in Figure 11. It could be observed that the location errors were 5 mm, 4 mm and 7.07 mm, respectively. The periods of time required to calculate the damage factor are shown in Figure 12.

It could be seen that the positioning accuracy of these three algorithms were about of the same level, but the technique put forward in this study has less computing time, which shows that the method is both simple and efficient.

## 6. Conclusions

A technique of damage location imaging for composite plates was put forward on the basic of difference signals and Lamb wave tomography. Followed by the calculation of the damage factor by the means of quadratic sum of the difference signal, and the location of damage was determined by the algorithm of probabilistic imaging. The main conclusions obtained are as follows:(1)In terms of the Lamb wave tomography, only the study on the difference of signals before and after damage are required so as to identify the location of damage. Thus, the inaccurate location of composite materials caused by the existence of wave velocity could be avoided.(2)There are differences in the direct wave signals of the response signals before and after the damage of the composite structure. The damage can be effectively located by the calculation method for damage factor based on quadratic sum of the difference signals.(3)Through simulation and experiment, the damage imaging of composite materials is realized. It is obtained that the maximum positioning error is 7.07 mm. The relative error is 1.67%. Moreover, the proposed method has the characteristics of simple calculation and high efficiency.

The method put forward in this study can effectively recognize the location of the damage in composite materials, with more extensive application potential in the assessment of damage for composite materials.

## Figures and Tables

**Figure 1 materials-13-00218-f001:**
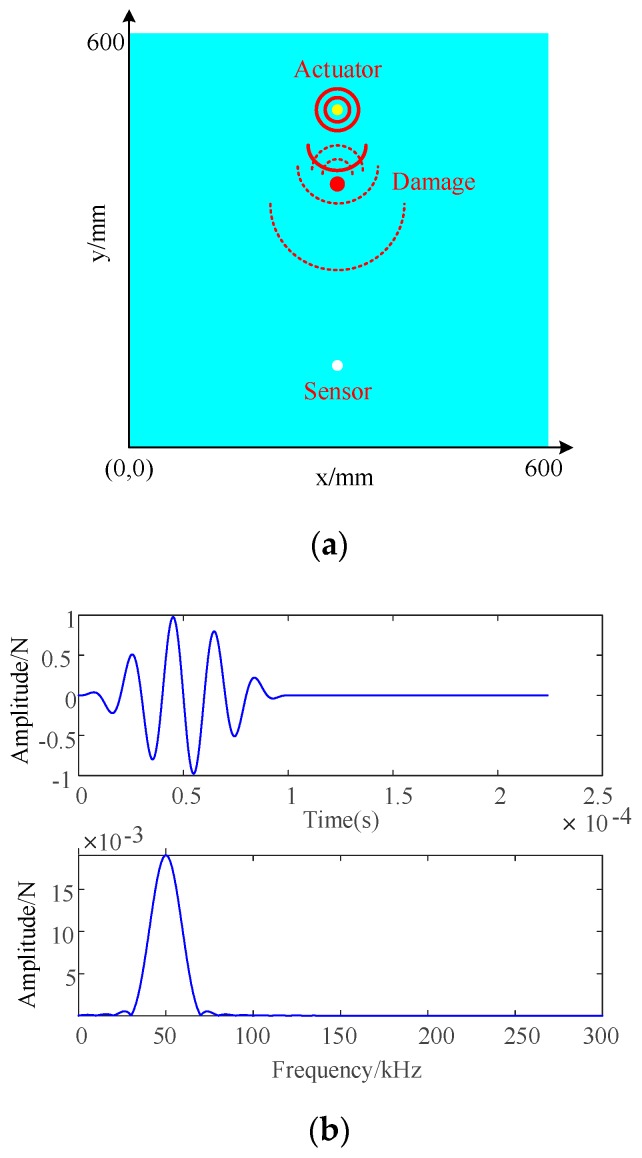
(**a**) Schematic diagram of actuator and sensor position; (**b**) high frequency excitation signal.

**Figure 2 materials-13-00218-f002:**
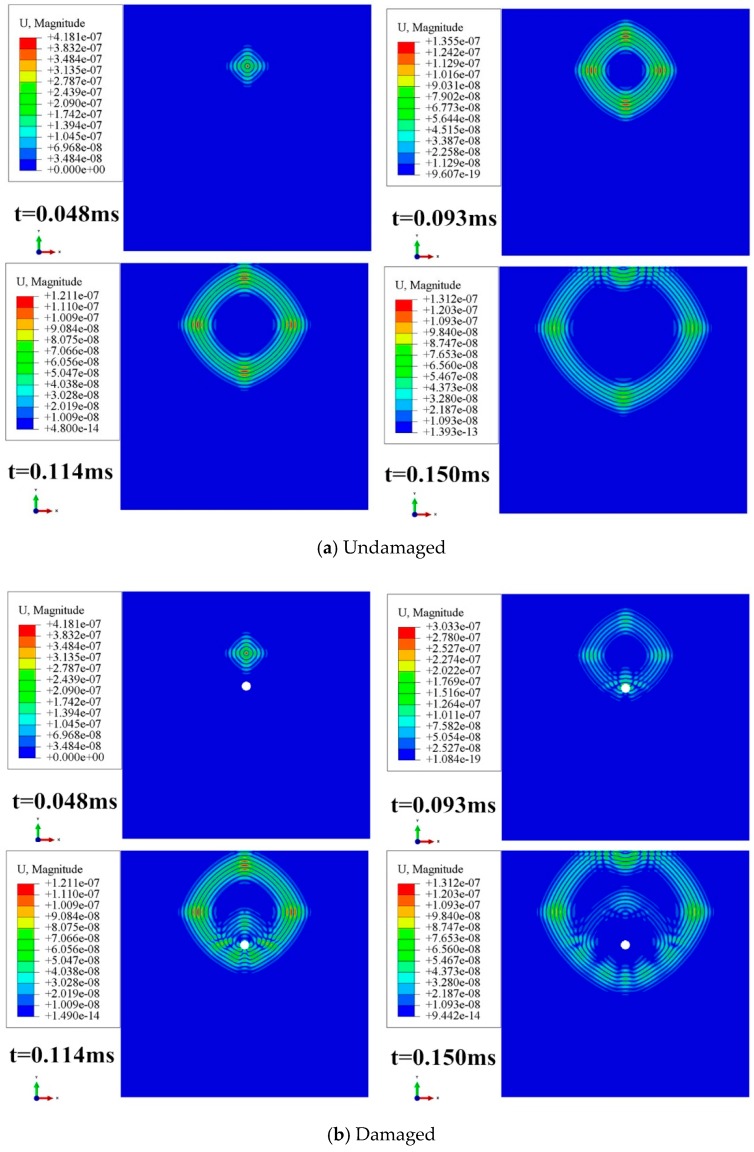
Lamb wave propagation cloud map in carbon fiber reinforced polymer (CFRP).

**Figure 3 materials-13-00218-f003:**
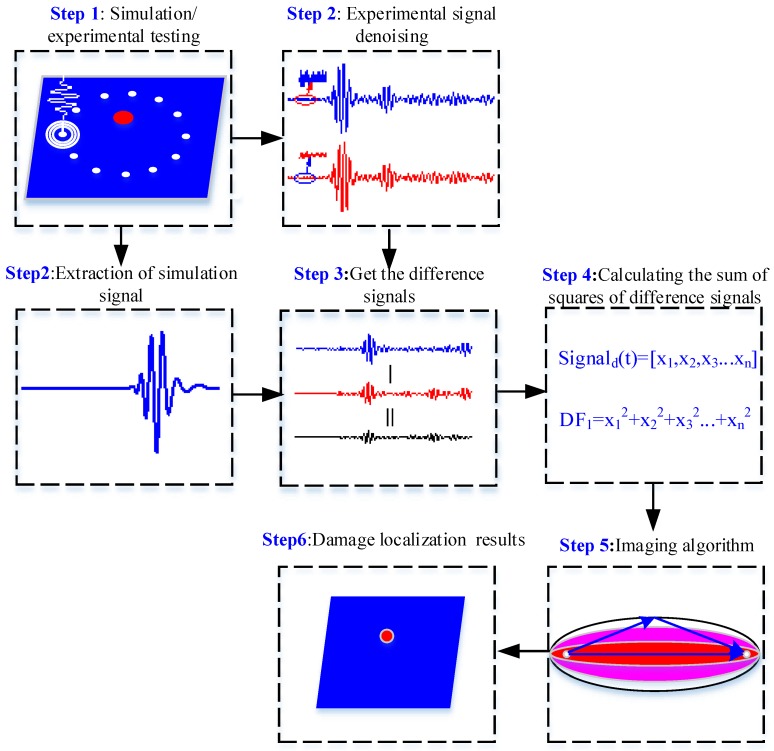
Flow chart of composite damage imaging.

**Figure 4 materials-13-00218-f004:**
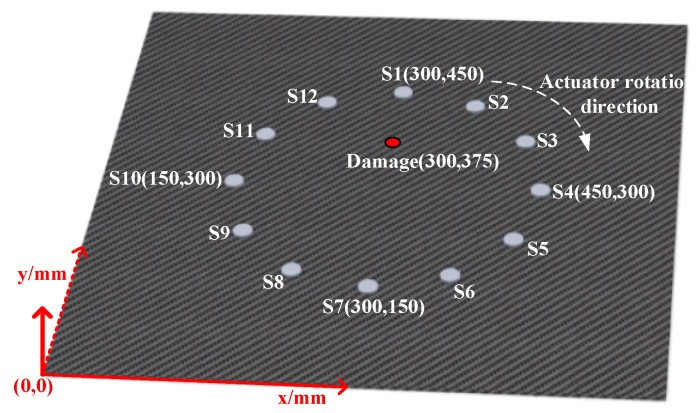
Schematic diagram of sensor and damage location.

**Figure 5 materials-13-00218-f005:**
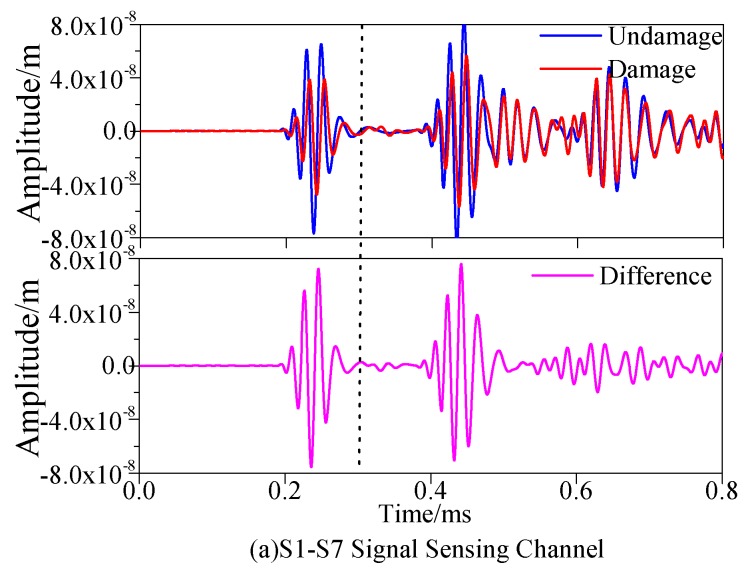
Response signals with different sensing channels. (**a**) S1–S7 Signal Sensing Channel (**b**) S5–S9 Signal Sensing Channel.

**Figure 6 materials-13-00218-f006:**
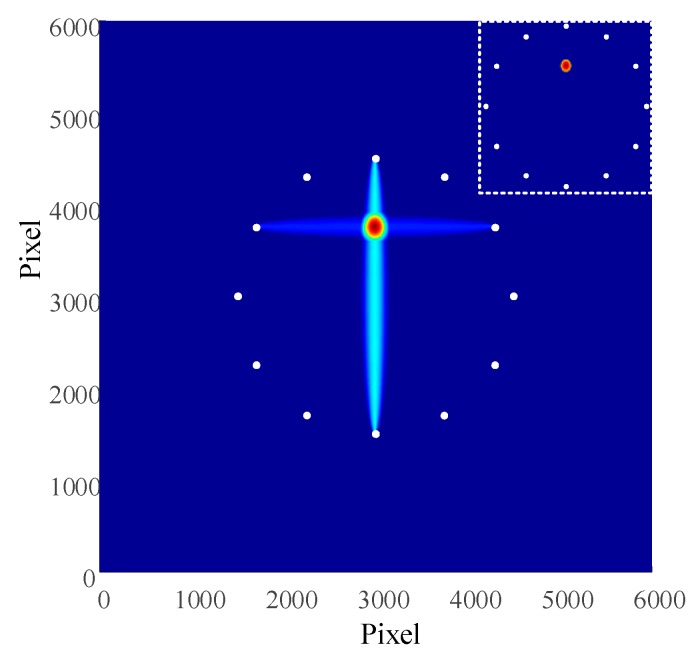
Damage imaging results.

**Figure 7 materials-13-00218-f007:**
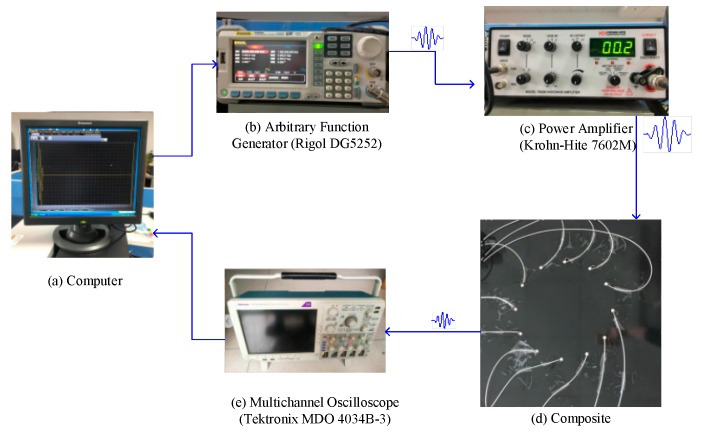
Photographs for the experimental system of Lamb wave tomography.

**Figure 8 materials-13-00218-f008:**
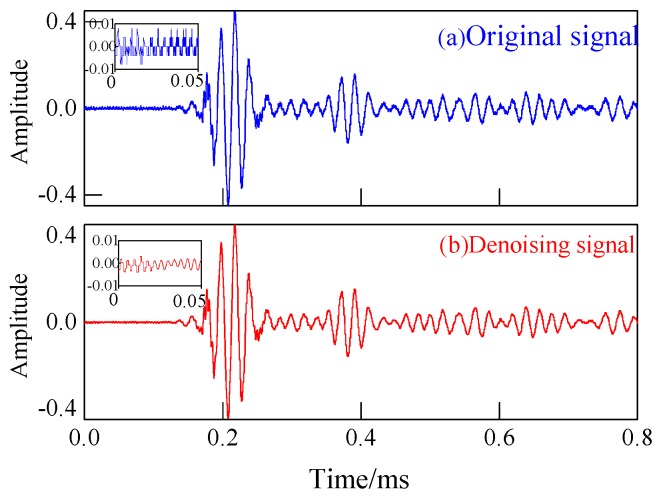
The original signal and denoising signal.

**Figure 9 materials-13-00218-f009:**
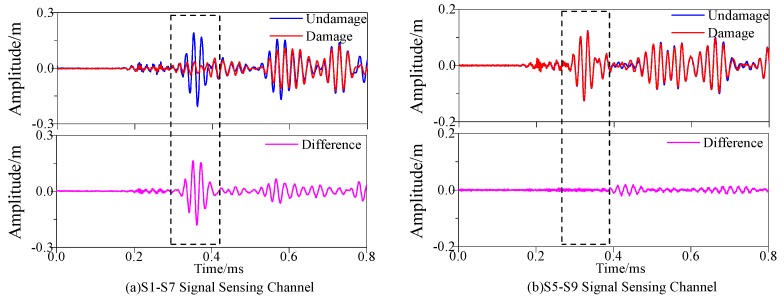
Response signals with different sensing channels. (**a**) S1–S7 Signal Sensing Channel (**b**) S5–S9 Signal Sensing Channel.

**Figure 10 materials-13-00218-f010:**
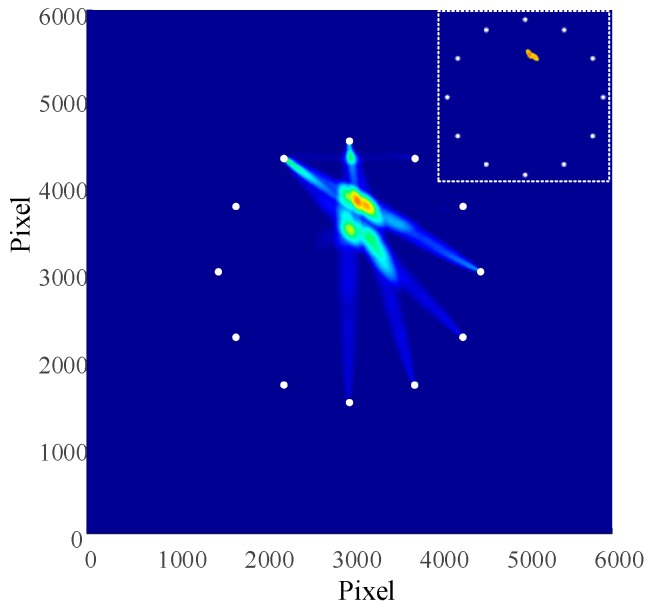
Damage imaging results.

**Figure 11 materials-13-00218-f011:**
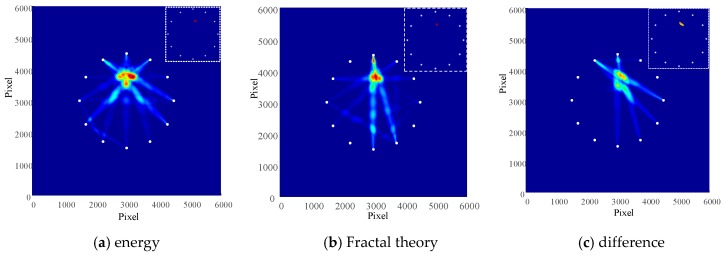
Damage imaging location results of different algorithms. (**a**) energy; (**b**) Fractal theory; (**c**) difference.

**Figure 12 materials-13-00218-f012:**
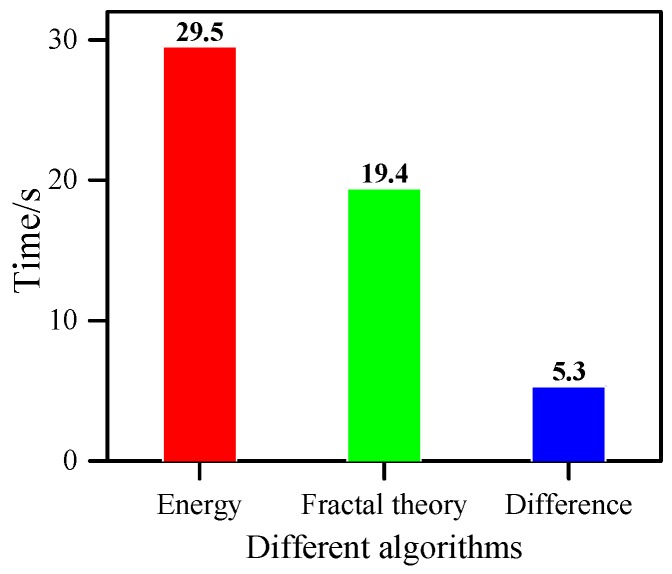
Time required to calculate damage factor by different algorithms.

**Table 1 materials-13-00218-t001:** Mechanical parameters of composite.

Elastic Properties		Strength		Fracture Energy		Density
E_1_	110 GPa	X^T^	2093 MPa	G_ft_	10 N/mm	1700 kg/m^3^
E_1_	7.8 GPa	X^C^	870 MPa	G_fc_	10 N/mm	-
*ν* _12_	0.32	Y^T^	50 MPa	G_mt_	1 N/mm	-
G_12_	40 GPa	Y^C^	198 MPa	G_mc_	1 N/mm	-
G_13_	40 GPa	S^L^	104 MPa	-	-	-
G_23_	40 GPa	-	-	-	-	-

E_1_ and E_2_—modulus of elasticity; G_12_, G_13_ and G_23_—shear modulus; G_ft_, G_fc_, G_mt_ and G_mc_—fracture energy; *ν*_12_—Poisson ratio; X^T^—tensile strength in the fiber direction; X^C^—compressive strength in the fiber direction; Y^T^—tensile strength in the transverse direction; Y^C^—compressive strength in the transverse direction; S^L^—shear strength in the fiber and transverse plane.

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
