# Peer review of "Damage Localization of Composites Based on Difference Signal and Lamb Wave Tomography"

_materials, 2020, doi:10.3390/ma13010218_

Round 1

Reviewer 1 Report

Thank You for the manuscript.

The paper claims to study the damage localization of composites based on different signal and Lamb wave tomography. The research presents a new technique that seems to be simple and efficient in comparison with existing methods. The work can be published because it is well readable and it presents useful research results achieved by both experimental study and simulation. However, the manuscript needs major corrections. Below are the aspects that need clarification or correction:

What should be the procedure before the placement of the piezoelectric sensors in a real situation when a position of the damage is difficult to suppose? (= What is necessary to do to be damage localized in an area that is small/large enough to be the sensors placed correctly?) How big the area - in maximum - can be investigated by using the presented method during one positioning of sensors? (= How big could be the diameter of the investigated array?) Will be the error bigger or the same if the array diameter at measurement will be more then 30 cm? How is the reliability of the achieved results at the new method – how many repetitions in experimental measurements have been done? Variables in the equations and descriptions in the text are not written in the same way (in the text should be in italic, they differ also in lower and upper letters) Units are written not correctly (e.g. khz), a blank space should be between a number and unit. The values for location errors differ in the text (in the line e.g. 248 and in Abstract it is stated 7.07 mm, and in the Conclusions, it is 7.7 mm).

Author Response

I would like to thank the expert for their valuable comments and revise the papers carefully according to their comments. Please see the attachment.

Reviewer 2 Report

Line 79. What does the number 8 after [0 ° / 90 °] mean? Table 1. What are the trade names of these materials? The numerical model is not sufficiently described? E.g. What type of material model was used isotropic, anisotropic, orthotropic? I do not know.
Figure 1 is missing amplitude units. Formula (1) no units. Unify the size of the describing letters of refers to the frequency fc.
What do the scales describe in Fig. 2? In general, this article is very interesting.Greetings.

Author Response

(The authors gave the same response as above.)

Reviewer 3 Report

The manuscript by Su et.al, is a numerical and experimental effort for identifying the position of localized damage in a CFRP.  The authors use both simulation and experiments to shows that using difference signal algorithm is fast an accurate algorithm for detecting damage location in CFRP sample.  

The paper is fairly good written. The introduction have to be improved to include more details on why other methods are not computationally efficiency and what is the consequence of that. Also, reporting that maximum positioning error is 7.7 mm in abstract and conclusion doesn’t make any sense. How much is the percentage error? 20% 30% 50% with respect to actual?

Some specific comments to address:

1- The introduction could be improved by mentioning broader application of methods. For example, sentences like the following can be added to the introduction line 59.

“Damage indication based on a single damage index value demonstrated its advantages in other methods such as vibro-acoustic modulation technique which facilitates interpretation of damage in the structures as well as monitoring of the damage evolution [1][2], [3].”

[1]         D. M. Donskoy and M. Ramezani, “Separation of amplitude and frequency modulations in Vibro-Acoustic Modulation Nondestructive Testing Method,” in Proceedings of Meetings on Acoustics, 2018, vol. 34, no. 1, p. 045002.

[2]         M. G. Ramezani, B. Golchinfar, D. Donskoy, S. Hassiotis, and G. Venkiteela, “Steel Material Degradation Assessment Via Vibro-Acoustic Modulation Technique,” Transp. Res. Rec. J. Transp. Res. Board, p. 036119811983827, May 2019.

[3] Schabowicz, Krzysztof. "Non-Destructive Testing of Materials in Civil Engineering." (2019): 3237.

2- Line 38. You used TOF before its definition in line 39. time-of-flight (TOF).

3- Line 67, “However, in these methods, the calculation of damage index involves complex mathematical problems and is inefficient, which is not conducive to rapid imaging of structural damage.” Is there any reference for this?

4- How did author generate the damaged signal from the simulation?

5-  Please add more explanation on Energy and fractal theory and what is the main difference between them and the difference method.

6- Please explain why other methods are more expensive than difference method.  

Author Response

(The authors gave the same response as above.)

Round 2

Reviewer 1 Report

In my opinion, the authors responded adequately and sufficiently.

Reviewer 3 Report

The authors have answered my comments accordingly and I agree on that publication of the paper in the present form.